# How to train your ViT? Data, Augmentation, and Regularization in Vision Transformers

**Andreas Steiner**[*]                                      *andstein@google.com*

**Alexander Kolesnikov**[*]                                 *akolesnikov@google.com*

**Xiaohua Zhai**[*]                                         *xzhai@google.com*

**Ross Wightman**[†]                                        *rwightman@gmail.com*

**Jakob Uszkoreit**                                         *usz@google.com*

**Lucas Beyer**[*]                                          *lbeyer@google.com*

*Google Research, Brain Team, Zürich*

[*] *Equal technical contribution,* [†] *independent researcher*

**Reviewed on OpenReview:** *https://openreview.net/forum?id=4nPswr1KcP*

## Abstract

Vision Transformers (ViT) have been shown to attain highly competitive performance for a wide range of vision applications, such as image classification, object detection and semantic image segmentation. In comparison to convolutional neural networks, the Vision Transformer's weaker inductive bias is generally found to cause an increased reliance on model regularization or data augmentation ("AugReg" for short) when training on smaller training datasets. We conduct a systematic empirical study in order to better understand the interplay between the amount of training data, AugReg, model size and compute budget.[1] As one result of this study we find that the combination of increased compute and AugReg can yield models with the same performance as models trained on an order of magnitude more training data: we train ViT models of various sizes on the public ImageNet-21k dataset which either match or outperform their counterparts trained on the larger, but not publicly available JFT-300M dataset.

## 1 Introduction

The Vision Transformer (ViT) (13) has recently emerged as a competitive alternative to convolutional neural networks (CNNs) that are ubiquitous across the field of computer vision. Without the translational equivariance of CNNs, ViT models are generally found to perform best in settings with large amounts of training data (13) or to require strong AugReg schemes to avoid overfitting (39). However, so far there was no comprehensive study of the trade-offs between model regularization, data augmentation, training data size and compute budget in Vision Transformers.

In this work, we fill this knowledge gap by conducting a thorough empirical study. We pre-train a large collection of ViT models (different sizes and hybrids with ResNets (18)) on datasets of different sizes, while

---

[1]We release more than 50 000 ViT models trained under diverse settings on various datasets. We believe this to be a treasure trove for model analysis. Available at https://github.com/google-research/vision_transformer and https://github.com/rwightman/pytorch-image-models. The code for full reproduction of model training is available at https://github.com/google-research/big_vision.

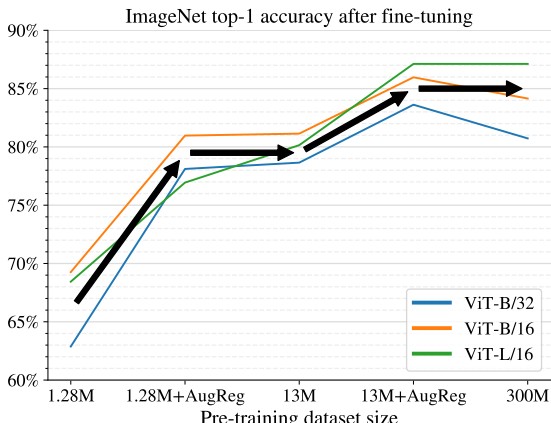

Figure 1: Adding the right amount of regularization and image augmentation can lead to similar gains as increasing the dataset size by an order of magnitude.

at the same time performing carefully designed comparisons across different amounts of regularization and data augmentation. We then proceed with extensive transfer learning experiments for the resulting models. We focus mainly on gaining insights useful for a practitioner with limited compute and data budgets.

The homogeneity of the performed study constitutes one of the key contributions of this paper. For the vast majority of works involving Vision Transformers it is not practical to retrain all baselines and proposed methods on equal footing, in particular those trained on larger amounts of data. Furthermore, there are numerous subtle and implicit design choices that cannot be controlled for effectively, such as the precise implementation of complex augmentation schemes, hyper-parameters (e.g. learning rate schedule, weight decay), test-time preprocessing, dataset splits and so forth. Such inconsistencies can result in significant amounts of noise added to the results, quite possibly affecting the ability to draw any conclusions. Hence, all models on which this work reports have been trained and evaluated in a consistent setup.

The insights we draw from our study constitute another important contribution of this paper. In particular, we demonstrate that carefully selected regularization and augmentations roughly correspond (from the perspective of model accuracy) to a 10x increase in training data size. However, regardless of whether the models are trained with more data or better AugRegs, one has to spend roughly the same amount of compute to get models attaining similar performance. We further evaluate if there is a difference between adding data or better AugReg when fine-tuning the resulting models on datasets of various categories. Other findings, such as the overall beneficial effect of AugRegs for medium-sized datasets, simply confirm commonly held beliefs. For those findings, the value of this study lies not in novelty, but rather in confirming these assumptions and quantifying their effect in a strictly controlled setting.

In addition, we aim to shed light on other aspects of using Vision Transformers in practice such as comparing transfer learning and training from scratch for mid-sized datasets. Finally, we evaluate various compute versus performance trade-offs. We discuss all of the aforementioned insights and more in detail in Section 4.

## 2 Scope of the study

With the ubiquity of modern deep learning (25) in computer vision it has quickly become common practice to pre-train models on large datasets once and re-use their parameters as initialization or feature extraction part in models trained on a broad variety of other tasks (32; 45).

In this setup, there are multiple ways to characterize computational and sample efficiency. When simply considering the overall costs of pre-training and subsequent training or fine-tuning procedures together, the cost of pre-training usually dominates, often by orders of magnitude. From the vantage point of a researcher aiming to improve model architectures or pre-training schemes, the pre-training costs might therefore be most

relevant. Most practitioners, however, rarely, if ever perform pre-training on today's largest datasets but instead use some of the many publicly available parameter sets. For them the costs of fine-tuning, adaptation or training a task-specific model from scratch would be of most interest.

Yet another valid perspective is that all training costs are effectively negligible since they are amortized over the course of the deployment of a model in applications requiring a very large number of invocations of inference.

In this setup there are different viewpoints on computational and data efficiency aspects. One approach is to look at the overall computational and sample cost of both pre-training and fine-tuning. Normally, "pre-training cost" will dominate overall costs. This interpretation is valid in specific scenarios, especially when pre-training needs to be done repeatedly or reproduced for academic/industrial purposes. However, in the majority of cases the pre-trained model can be downloaded or, in the worst case, trained once in a while. Contrary, in these cases, the budget required for adapting this model may become the main bottleneck.

Thus, we pay extra attention to the scenario, where the cost of obtaining a pre-trained model is free or effectively amortized by future adaptation runs. Instead, we concentrate on time and compute spent on finding a good adaptation strategy (or on tuning from scratch training setup), which we call "practitioner's cost".

A more extreme viewpoint is that the training cost is not crucial, and all that matters is eventual inference cost of the trained model, "deployment cost", which will amortize all other costs. This is especially true for large scale deployments, where a visual model is expected to be used a massive number of times. Overall, there are three major viewpoints on what is considered to be the central cost of training a vision model. In this study we touch on all three of them, but mostly concentrate on "practitioner" and "deployment" costs.

## 3 Experimental setup

In this section we describe our unified experimental setup, which is used throughout the paper. We use a single JAX/Flax (19; 3) codebase for pre-training and transfer learning using TPUs. Inference speed measurements, however, were obtained on V100 GPUs (16G) using the `timm` PyTorch library (42). All datasets are accessed through the *TensorFlow Datasets* library (15), which helps to ensure consistency and reproducibility. More details of our setup are provided below.

### 3.1 Datasets and metrics

For pre-training we use two large-scale image datasets: ILSVRC-2012 (ImageNet-1k) and ImageNet-21k. ImageNet-21k dataset contains approximately 14 million images with about 21 000 distinct object categories (11; 22; 30). ImageNet-1k is a subset of ImageNet-21k consisting of about 1.3 million training images and 1000 object categories. We make sure to de-duplicate images in ImageNet-21k with respect to the test sets of the downstream tasks as described in (13; 22). Additionally, we used ImageNetV2 (29) for evaluation purposes.

For transfer learning evaluation we use 4 popular computer vision datasets from the VTAB benchmark (45): CIFAR-100 (24), Oxford IIIT Pets (28) (or Pets37 for short), Resisc45 (6) and Kitti-distance (14). We selected these datasets to cover the standard setting of natural image classification (CIFAR-100 and Pets37), as well as classification of images captured by specialized equipment (Resisc45) and geometric tasks (Kitti-distance). In some cases we also use the full VTAB benchmark (19 datasets) to additionally ensure robustness of our findings.

For all datasets we report top-1 classification accuracy as our main metric. Hyper-parameters for fine-tuning are selected by the result from the *validation* split, and final numbers are reported from the *test* split. Note that for ImageNet-1k we follow common practice of reporting our main results on the validation set. Thus, we set aside 1% of the training data into a *minival* split that we use for model selection. Similarly, we use a minival split for CIFAR-100 (2% of training split) and Oxford IIIT Pets (10% of training split). For Resisc45, we use only 60% of the training split for training, and another 20% for validation, and 20% for computing test metrics. Kitti-distance finally comes with an official validation and test split that we use for the intended purpose. See (45) for details about the VTAB dataset splits.

Table 1: Configurations of ViT models.

| Model | Layers | Width | MLP | Heads | Params |
|-------|--------|-------|------|-------|--------|
| ViT-Ti (39) | 12 | 192 | 768 | 3 | 5.8M |
| ViT-S (39) | 12 | 384 | 1536 | 6 | 22.2M |
| ViT-B (13) | 12 | 768 | 3072 | 12 | 86M |
| ViT-L (13) | 24 | 1024 | 4096 | 16 | 307M |

Table 2: ResNet+ViT hybrid models.

| Model | Resblocks | Patch-size | Params |
|-------|-----------|------------|--------|
| R+Ti/16 | [] | 8 | 6.4M |
| R26+S/32 | [2, 2, 2, 2] | 1 | 36.6M |
| R50+L/32 | [3, 4, 6, 3] | 1 | 330.0M |

## 3.2 Models

This study focuses mainly on the Vision Transformer (ViT) (13). We use 4 different configurations from (13; 39): ViT-Ti, ViT-S, ViT-B and ViT-L, which span a wide range of different capacities. The details of each configuration are provided in Table 1. We use patch-size 16 for all models, and additionally patch-size 32 for the ViT-S and ViT-B variants. The only difference to the original ViT model (13) in our paper is that we drop the hidden layer in the head, as empirically it does not lead to more accurate models and often results in optimization instabilities: when pre-training on ImageNet-1k we include both models with and without hidden layer, when pre-training on ImageNet-21k we always drop the hidden layer.

In addition, we train hybrid models that first process images with a ResNet (18) backbone and then feed the spatial output to a ViT as the initial patch embeddings. We use a ResNet stem block ($7 \times 7$ convolution + batch normalization + ReLU + max pooling) followed by a variable number of bottleneck blocks (18). We use the notation R$n$+{Ti,S,L}/$p$ where $n$ counts the number of convolutions, and $p$ denotes the patch-size *in the input image* - for example R+Ti/16 reduces image dimensions by a factor of two in the ResNet stem and then forms patches of size 8 as an input to the ViT, which results in an effective patch-size of 16.

## 3.3 Regularization and data augmentations

To regularize our models we use robust regularization techniques widely adopted in the computer vision community. We apply dropout to intermediate activations of ViT as in (13). Moreover, we use the stochastic depth regularization technique (20) with linearly increasing probability of dropping layers.

For data augmentation, we rely on the combination of two recent techniques, namely Mixup (47) and RandAugment (7). For Mixup, we vary its parameter $\alpha$, where 0 corresponds to no Mixup. For RandAugment, we vary the magnitude parameter $m$, and the number of augmentation layers $l$. Note that we use the original RandAugment implementation in TensorFlow, which differs from re-implementations found, for example, in timm (42).

We also try two values for weight decay (27) which we found to work well, since increasing AugReg may need a decrease in weight decay (2).

Overall, our sweep contains 28 configurations, which is a cross-product of the following hyper-parameter choices:

- Either use no dropout and no stochastic depth (e.g. no regularization) or use dropout with probability 0.1 and stochastic depth with maximal layer dropping probability of 0.1, thus 2 configuration in total.

- 7 data augmentation setups for $(l, m, \alpha)$: none $(0, 0, 0)$, light1 $(2, 0, 0)$, light2 $(2, 10, 0.2)$, medium1 $(2, 15, 0.2)$, medium2 $(2, 15, 0.5)$, strong1 $(2, 20, 0.5)$, strong2 $(2, 20, 0.8)$.

- Weight decay: 0.1 or 0.03. The weight decay is decoupled following (27), but multiplied by the learning-rate which peaks at 0.001.

## 3.4 Pre-training

We pre-trained the models with Adam (21), using $\beta_1 = 0.9$ and $\beta_2 = 0.999$, with a batch size of 4096, and a cosine learning rate schedule with a linear warmup (10k steps). To stabilize training, gradients were clipped at global norm 1. The images are pre-processed by Inception-style cropping (36) and random horizontal

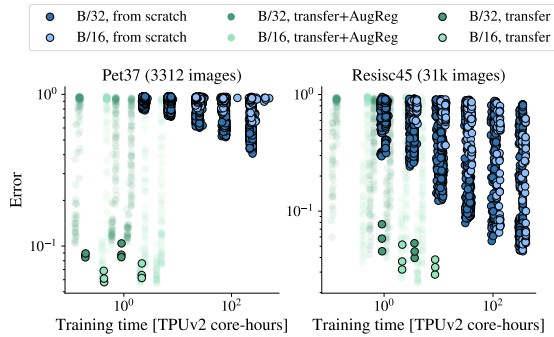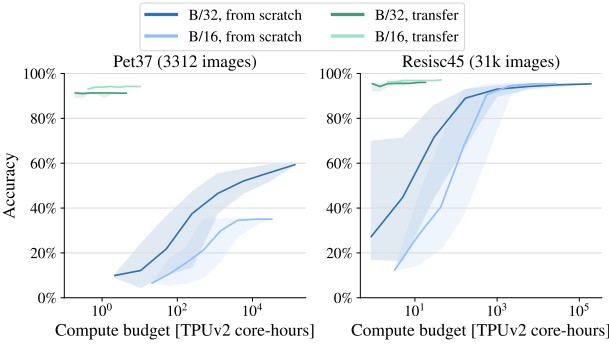

Figure 2: **Left**: When training small and mid-sized datasets from scratch it is very hard to achieve a test error that can trivially be attained by fine-tuning a model pre-trained on a large dataset like ImageNet-21k. With our recommended models (Section 4.5), one can find a good solution with very few trials (bordered green dots, using recipe from B). Note that AugReg is not helpful when transferring pre-trained models (borderless green dots). **Right**: Same data as on the left side (ignoring the borderless green dots), but simulating the results of a random search. For a given compute budget (x-axis), choosing random configurations within that budget leads to varying final performance, depending on choice of hyper parameters (shaded area covers 90% from 1000 random samples, line corresponds to median).

flipping. On the smaller ImageNet-1k dataset we trained for 300 epochs, and for 30 and 300 epochs on the ImageNet-21k dataset. Since ImageNet-21k is about 10x larger than ImageNet-1k, this allows us to examine the effects of the increased dataset size also with a roughly constant total compute used for pre-training.

### 3.5 Fine-tuning

We fine-tune with SGD with a momentum of 0.9 (storing internal state as `bfloat16`), sweeping over 2-3 learning rates and 1-2 training durations per dataset as detailed in Table 4 in the appendix. We used a fixed batch size of 512, gradient clipping at global norm 1 and a cosine decay learning rate schedule with linear warmup. Fine-tuning was done both at the original resolution (224), as well as at a higher resolution (384) as described in (40).

## 4 Findings

### 4.1 Scaling datasets with AugReg and compute

One major finding of our study, which is depicted in Figure 1, is that by judicious use of image augmentations and model regularization, one can (pre-)train a model to similar accuracy as by increasing the dataset size by about an order of magnitude. More precisely, our best models trained on AugReg ImageNet-1k (31) perform about equal to the same models pre-trained on the 10x larger plain ImageNet-21k (11) dataset. Similarly, our best models trained on AugReg ImageNet-21k, when compute is also increased (e.g. training run longer), match or outperform those from (13) which were trained on the plain JFT-300M (35) dataset with 25x more images. Thus, it is possible to *match these private results with a publicly available dataset*, and it is imaginable that training longer and with AugReg on JFT-300M might further increase performance.

Of course, these results cannot hold for arbitrarily small datasets. For instance, according to Table 5 of (44), training a ResNet50 on only 10% of ImageNet-1k with heavy data augmentation improves results, but does not recover training on the full dataset.

### 4.2 Transfer is the better option

Here, we investigate whether, for reasonably-sized datasets a practitioner might encounter, it is advisable to try training from scratch with AugReg, or whether time and money is better spent transferring pre-trained

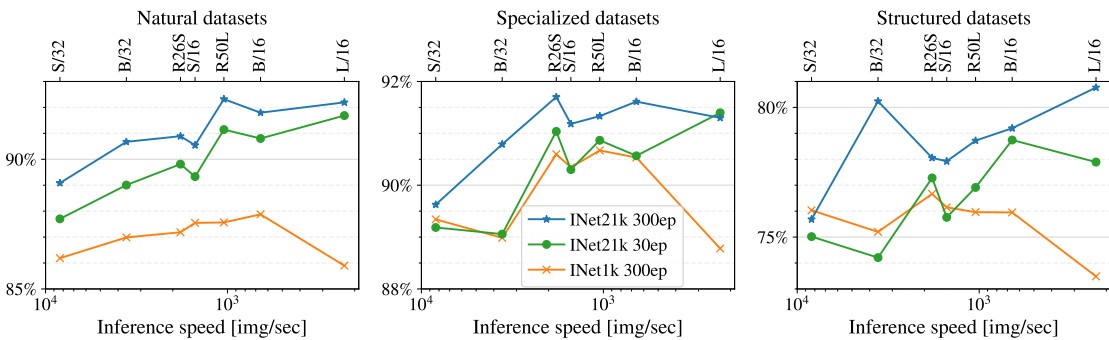

Figure 3: Pretraining on more data yields more transferable models on average, tested on the VTAB suite (45) of 19 tasks across 3 categories.

models that are freely available. The result is that, for most practical purposes, transferring a pre-trained model is both more cost-efficient and leads to better results.

We perform a thorough search for a good training recipe[2] for both the small ViT-B/32 and the larger ViT-B/16 models on two datasets of practical size: Pet37 contains only about 3000 training images and is relatively similar to the ImageNet-1k dataset. Resisc45 contains about 30 000 training images and consists of a very different modality of satellite images, which is not well covered by either ImageNet-1k or ImageNet-21k. Figure 2 shows the result of this search.

The most striking finding is that, no matter how much training time is spent, for the tiny Pet37 dataset, it does not seem possible to train ViT models from scratch to reach accuracy anywhere near that of transferred models. Furthermore, since pre-trained models are freely available for download, the pre-training cost for a practitioner is effectively zero, only the compute spent on transfer matters, and thus *transferring a pre-trained model is simultaneously significantly cheaper and gives better results.*

For the larger Resisc45 dataset, this result still holds, although spending two orders of magnitude more compute and performing a heavy search may come close (but not reach) to the accuracy of pre-trained models.

Notably, this does not account for the "exploration cost", which is difficult to quantify. For the pre-trained models, we highlight those which performed best on the *pre-training* validation set and could be called *recommended models* (see Section 4.5). We can see that using a recommended model has a high likelihood of leading to good results in just a few attempts, while this is not the case for training from-scratch, as evidenced by the wide vertical spread of points.

### 4.3   More data yields more generic models

We investigate the impact of pre-training dataset size by transferring pre-trained models to unseen downstream tasks. We evaluate the pre-trained models on VTAB, including 19 diverse tasks (45).

Figure 3 shows the results on three VTAB categories: natural, specialized and structured. The models are sorted by the inference time per step, thus the larger model the slower inference speed. We first compare two models using the same compute budget, with the only difference being the dataset size of ImageNet-1k (1.3M images) and ImageNet-21k (13M images). We pre-train for 300 epochs on ImageNet-1k, and 30 epochs on ImageNet-21k. Interestingly, the model pre-trained on ImageNet-21k is significantly better than the ImageNet-1k one, across all the three VTAB categories. This is in contrast with the validation performance on ImageNet-1k (Figure 6), where this difference does not appear so clearly.

As the compute budget keeps growing, we observe consistent improvements on ImageNet-21k dataset with 10x longer schedule. On a few almost solved tasks, e.g. flowers, the gain is small in absolute numbers. For

---

[2]Not only do we further increase available AugReg settings, but we also sweep over other generally important training hyperparameters: learning-rate, weight-decay, and training duration, as described in Appendix A.

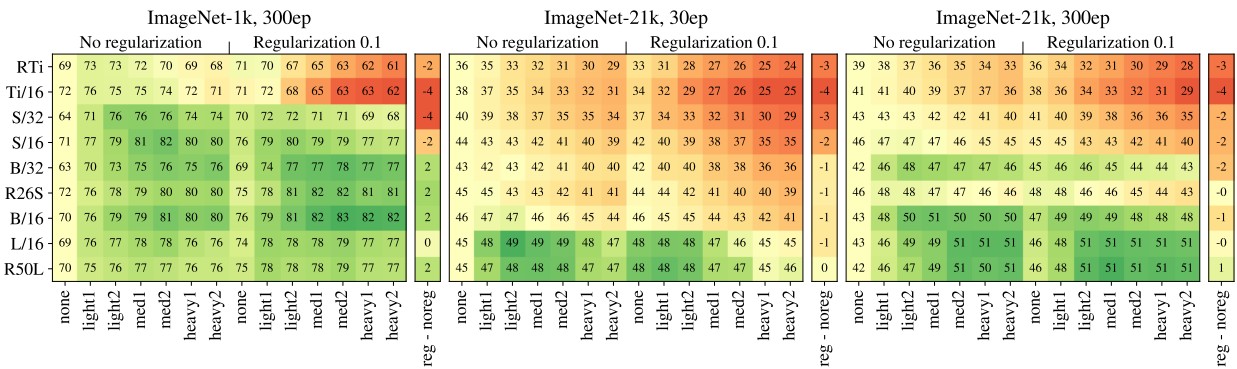

Figure 4: Validation accuracy (for ImageNet-1k: minival accuracy) when using various amounts of augmentation and regularization, highlighting differences to the unregularized, unaugmented setting. For relatively small amount of data, almost everything helps. However, when switching to ImageNet-21k while keeping the training budget fixed, almost everything hurts; only when also increasing compute, does AugReg help again. The single column right of each plot show the difference between the best setting with regularization and the best setting without, highlighting that regularization typically hurts on ImageNet-21k.

the rest of the tasks, the improvements are significant compared to the model pre-trained for a short schedule. All the detailed results on VTAB could be found from supplementary section C.

Overall, we conclude that *more data yields more generic models*, the trend holds across very diverse tasks. We recommend the design choice of using *more data* with a fixed compute budget.

## 4.4 Prefer augmentation to regularization

It is not clear a priori what the trade-offs are between data augmentation such as RandAugment and Mixup, and model regularization such as Dropout and StochasticDepth. In this section, we aim to discover general patterns for these that can be used as rules of thumb when applying Vision Transformers to a new task. In Figure 4, we show the upstream validation score obtained for each individual setting, i.e. numbers are not comparable when changing dataset. The colour of a cell encodes its improvement or deterioration in score when compared to the unregularized, unaugmented setting, i.e. the leftmost column. Augmentation strength increases from left to right, and model "capacity" increases from top to bottom.

The first observation that becomes visible, is that for the mid-sized ImageNet-1k dataset, any kind of AugReg helps. However, when using the 10x larger ImageNet-21k dataset and keeping compute fixed, i.e. running for 30 epochs, any kind of AugReg *hurts* performance for all but the largest models. It is only when also increasing the computation budget to 300 epochs that AugReg helps more models, although even then, it continues hurting the smaller ones. Generally speaking, there are significantly more cases where adding augmentation helps, than where adding regularization helps. More specifically, the thin columns right of each map in Figure 4 shows, for any given model, its best regularized score minus its best unregularized score. This view, which is expanded in Figure 7 in the Appendix, tells us that when using ImageNet-21k, regularization almost always hurts.

## 4.5 Choosing which pre-trained model to transfer

As we show above, when pre-training ViT models, various regularization and data augmentation settings result in models with drastically different performance. Then, from the practitioner's point of view, a natural question emerges: how to select a model for further adaption for an end application? One way is to run downstream adaptation for all available pre-trained models and then select the best performing model, based on the validation score on the downstream task of interest. This could be quite expensive in practice. Alternatively, one can select a single pre-trained model based on the upstream validation accuracy and then only use this model for adaptation, which is much cheaper.

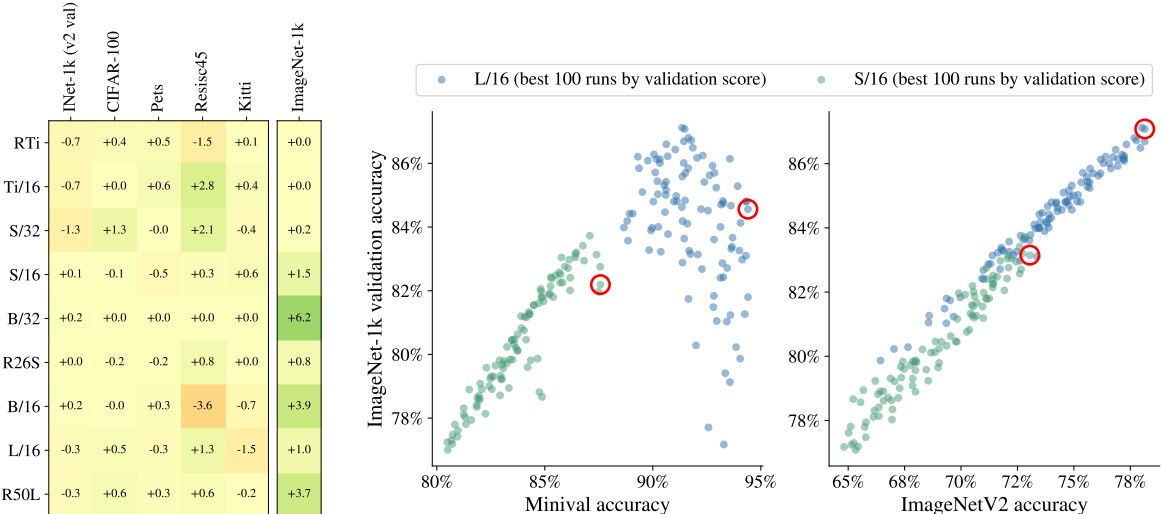

Figure 5: Choosing best models. **Left**: Difference of fine-tuning test scores between models chosen by best validation score on pre-training data vs. validation score on fine-tuning data (negative values mean that selecting models by pre-training validation deteriorates fine-tuning test metrics). **Right**: Correlation between "minival" validation score vs. ImageNetV2 validation score and official ImageNet-1k validation score (that serves as a test score in this study). Red circles highlight the best models by validation score, see Section 4.5 for an explanation.

In this section we analyze the trade-off between these two strategies. We compare them for a large collection of our pre-trained models on 5 different datasets. Specifically, in Figure 5 (left) we highlight the performance difference between the cheaper strategy of adapting only the best pre-trained model and the more expensive strategy of adapting *all* pre-trained models (and then selecting the best).

The results are mixed, but generally reflect that the cheaper strategy works equally well as the more expensive strategy in the majority of scenarios. Nevertheless, there are a few notable outliers, when it is beneficial to adapt all models. Thus, we conclude that selecting a single pre-trained model based on the upstream score is a cost-effective practical strategy and also use it throughout our paper. However, we also stress that if extra compute resources are available, then in certain cases one can further improve adaptation performance by fine-tuning additional pre-trained models.

**A note on validation data for the ImageNet-1k dataset.** While performing the above analysis, we observed a subtle, but severe issue with models pre-trained on ImageNet-21k and transferred to ImageNet-1k dataset. The validation score for these models (especially for large models) is not well correlated with observed test performance, see Figure 5 (right). This is due to the fact that ImageNet-21k data contains ImageNet-1k training data and we use a "minival" split from the training data for evaluation (see Section 3.1). As a result, large models on long training schedules memorize the data from the training set, which biases the evaluation metric computed in the "minival" evaluation set. To address this issue and enable fair hyper-parameter selection, we instead use the independently collected ImageNetV2 data (29) as the validation split for transferring to ImageNet-1k. As shown in Figure 5 (right), this resolves the issue. We did not observe similar issues for the other datasets. *We recommend that researchers transferring ImageNet-21k models to ImageNet-1k follow this strategy.*

### 4.6 Prefer increasing patch-size to shrinking model-size

One unexpected outcome of our study is that we trained several models that are roughly equal in terms of inference throughput, but vary widely in terms of their quality. Specifically, Figure 6 (right) shows that models containing the "Tiny" variants perform significantly worse than the similarly fast larger models with "/32" patch-size. For a given resolution, the patch-size influences the amount of tokens on which self-attention is

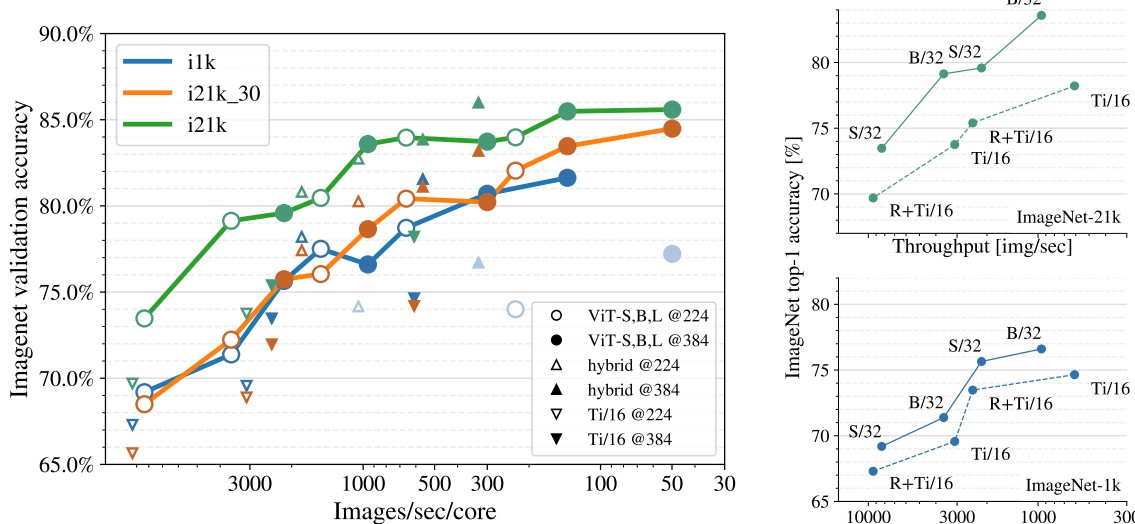

Figure 6: ImageNet transfer. **Left**: For every architecture and upstream dataset, we selected the best model by upstream validation accuracy. Main ViT-S,B,L models are connected with a solid line to highlight the trend, with the exception of ViT-L models pre-trained on i1k, where the trend breaks down. The same data is also shown in Table 3. **Right**: Focusing on small models, it is evident that using a larger patch-size (/32) significantly outperforms making the model thinner (Ti).

Table 3: ImageNet-1k transfer. Column $i1k_{up}$ evaluates best checkpoint without adaptation, columns $i1k_{300}$, $i21k_{30}$ and $i21k_{300}$ (ImageNet-1k 300 epochs and ImageNet-21k 30 and 300 epochs) report numbers after fine-tuning, which are shown in Figure 6, the "recommended checkpoints" (see Section 4.5) were fine-tuned with two different learning rates (see Section B). For the column $i21k^{v2}$ (ImageNet-21k, 300 epochs), the upstream checkpoint was instead chosen by ImageNetV2 validation accuracy. The JFT-300M numbers are taken from (13) (bold numbers indicate our results that are on par or surpass the published JFT-300M results without AugReg for the same models). Inference speed measurements were computed on an NVIDIA V100 GPU using `timm` (42), sweeping the batch size for best throughput.

| Model | 224px resolution | | | | | 384px resolution | | | | | |
|---|---|---|---|---|---|---|---|---|---|---|---|
| | img/sec | $i1k_{up}$ | $i1k_{300}$ | $i21k_{30}$ | $i21k_{300}$ | img/sec | $i1k_{300}$ | $i21k_{30}$ | $i21k_{300}$ | $i21k^{v2}$ | JFT300M |
| L/16 | 228 | 75.72 | 74.01 | 82.05 | 83.98 | 50 | 77.21 | 84.48 | 85.59 | 87.08 | **87.12** |
| B/16 | 659 | 79.84 | 78.73 | 80.42 | 83.96 | 138 | 81.63 | 83.46 | 85.49 | **86.15** | 84.15 |
| S/16 | 1508 | 79.00 | 77.51 | 76.04 | 80.46 | 300 | 80.70 | 80.22 | 83.73 | 83.15 | - |
| R50+L/32 | 1047 | 76.84 | 74.17 | 80.26 | 82.74 | 327 | 76.71 | 83.19 | 85.99 | 86.21 | - |
| R26+S/32 | 1814 | 79.61 | 78.20 | 77.42 | 80.81 | 560 | 81.55 | 81.11 | 83.85 | 83.80 | - |
| Ti/16 | 3097 | 72.59 | 69.56 | 68.89 | 73.75 | 610 | 74.64 | 74.20 | 78.22 | 77.83 | - |
| B/32 | 3597 | 74.42 | 71.38 | 72.24 | 79.13 | 955 | 76.60 | 78.65 | 83.59 | **83.59** | 80.73 |
| S/32 | 8342 | 72.07 | 69.19 | 68.49 | 73.47 | 2154 | 75.65 | 75.74 | 79.58 | 80.01 | - |
| R+Ti/16 | 9371 | 70.13 | 67.30 | 65.65 | 69.69 | 2426 | 73.48 | 71.97 | 75.40 | 75.33 | - |

performed and, thus, is a contributor to model capacity which is not reflected by parameter count. Parameter count is reflective neither of speed, nor of capacity (10).

## 5 Related work

The scope of this paper is limited to studying pre-training and transfer learning of Vision Transformer models and there already are a number of studies considering similar questions for convolutional neural networks (23; 22). Here we hence focus on related work involving ViT models.

As first proposed in (13), ViT achieved competitive performance only when trained on comparatively large amounts of training data, with state-of-the-art transfer results using the ImageNet-21k and JFT-300M datasets, with roughly 13M and 300M images, respectively. In stark contrast, (39) focused on tackling overfitting of ViT when training from scratch on ImageNet-1k by designing strong regularization and augmentation schemes. Yet neither work analyzed the effects of stronger augmentation of regularization and augmentation in the presence of larger amounts of training data.

Ever since (22) first showed good results when pre-training BiT on ImageNet-21k, more architecture works have mentioned using it for select few experiments (13; 38; 37; 8), with (30) arguing more directly for the use of ImageNet-21k. However, none of these works thoroughly investigates the combined use of AugReg and ImageNet-21k and provides conclusions, as we do here.

An orthogonal line of work introduces cleverly designed inductive biases in ViT variants or retain some of the general architectural parameters of successful convolutional architectures while adding self-attention to them. (33) carefully combines a standard convolutional backbone with bottleneck blocks based on self-attention instead of convolutions. In (26; 17; 41; 43) the authors propose hierarchical versions of ViT. (9) suggests a very elegant idea of initializing Vision Transformer, such that it behaves similarly to convolutional neural network in the beginning of training.

Yet another way to address overfitting and improve transfer performance is to rely on self-supervised learning objectives. (1) pre-trains ViT to reconstruct perturbed image patches. Alternatively, (4) devises a self-supervised training procedure based on the idea from (16), achieving impressive results. We leave the systematic comparison of self-supervised and supervised pre-training to future work.

## 6 Discussion

**Societal Impact.** Our experimental study is relatively thorough and used a lot of compute. This could be taken as encouraging anyone who uses ViTs to perform such large studies. On the contrary, our aim is to provide good starting points and off-the-shelf checkpoints that remove the need for such extensive search in future work.

**Limitations.** In order to be thorough, we restrict the study to the default ViT architecture and neither include ResNets, which have been well studied over the course of the past years, nor more recent ViT variants. We anticipate though that many of our findings extend to other ViT-based architectures as well.

# 7 Summary of recommendations

Below we summarize three main recommendations based on our study:

- We recommend to use checkpoints that were pre-trained on more upstream data, and not relying only on ImageNet-1k as a proxy for model quality, since ImageNet-1k validation accuracy is inflated when pre-training on ImageNet-1k, and more varied upstream data yields more widely applicable models (Figure 3 and Section 4.3).

- Judiciously applying data augmentation and model regularization makes it possible to train much better models on a dataset of a given size (Figure 1), and these improvements can be observed both with medium sized datasets like ImageNet-1k, and even with large datasets like ImageNet-21k. But there are no simple rules which AugReg settings to select. The best settings vary a lot depending on model capacity and training schedule, and one needs to be careful not to apply AugReg to a model that is too small, or when pre-training for too short – otherwise the model quality may deteriorate (see Figure 4 for an exhaustive quantitative evaluation and Section 4.4 for further comments on regularization vs augmentations).

- How to select the best upstream model for transfer on your own task? Aside from always using ImageNet-21k checkpoints, we recommend to select the model with the best upstream validation performance (Section 4.5, table with paths in our Github repository[3]). As we show in Figure 5, this choice is generally optimal for a wide range of tasks. If the user has additional computational resources available to fine-tune all checkpoints, they may get slightly better results in some scenarios, but also need to be careful with respect to ImageNet-1k and ImageNet-21k data overlap when it comes to model selection (Figure 5, right).

# 8 Conclusion

We conduct the first systematic, large scale study of the interplay between regularization, data augmentation, model size, and training data size when pre-training Vision Transformers, including their respective effects on the compute budget needed to achieve a certain level of performance. We also evaluate pre-trained models through the lens of transfer learning. As a result, we characterize a quite complex landscape of training settings for pre-training Vision Transformers across different model sizes. Our experiments yield a number of surprising insights around the impact of various techniques and the situations when augmentation and regularization are beneficial and when not.

We also perform an in-depth analysis of the transfer learning setting for Vision Transformers. We conclude that across a wide range of datasets, even if the downstream data of interest appears to only be weakly related to the data used for pre-training, transfer learning remains the best available option. Our analysis also suggests that among similarly performing pre-trained models, for transfer learning a model with more training data should likely be preferred over one with more data augmentation.

We hope that our study will help guide future research on Vision Transformers and will be a useful source of effective training settings for practitioners seeking to optimize their final model performance in the light of a given computational budget.

**Acknowledgements** We thank Alexey Dosovitskiy, Neil Houlsby, and Ting Chen for insightful feedback; the Google Brain team at large for providing a supportive research environment.

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

## A  From-scratch training details

We present from-scratch training details for B/32 and B/16 models, on both Resisc45 and Pets37 datasets. We perform a grid search over the following parameters:

- B/32 on Pets37
    - Epochs: $\{1\text{k}, 3\text{k}, 10\text{k}, 30\text{k}, 300\text{k}\}$
    - Learning rates: $\{1e-4, 3e-4, 1e-3, 3e-3\}$
    - Weight decays[4]: $\{1e-5, 3e-5, 1e-4, 3e-4\}$

- B/16 on Pets37
    - Epochs: $\{1\text{k}, 3\text{k}, 10\text{k}\}$
    - Learning rates: $\{3e-4, 1e-3\}$
    - Weight decays: $\{3e-5, 1e-4\}$

- B/32 on Resisc45
    - Epochs: $\{75, 250, 750, 2.5\text{k}, 7.5\text{k}, 25\text{k}\}$
    - Learning rates: $\{1e-4, 3e-4, 1e-3, 3e-3\}$
    - Weight decays: $\{1e-5, 3e-5, 1e-4, 3e-4\}$

- B/16 on Resisc45
    - Epochs: $\{75, 250, 750, 2.5\text{k}, 7.5\text{k}\}$
    - Learning rates: $\{1e-3\}$
    - Weight decays: $\{1e-4, 3e-4\}$

All these from-scratch runs sweep over dropout rate and stochastic depth in range $\{(0.0, 0.0), (0.1, 0.1), (0.2, 0.2)\}$, and data augmentation $(l, m, \alpha)$ in range $\{$ $(0, 0, 0)$, $(2, 10, 0.2)$, $(2, 15, 0.2)$, $(2, 15, 0.5)$, $(2, 20, 0.5)$, $(2, 20, 0.8)$, $(4, 15, 0.5)$, $(4, 20, 0.8)$ $\}$.

For the definition of $(l, m, \alpha)$ refer to Section 3.3

## B  Finetune details

In Table 4, we show the hyperparameter sweep range for finetune jobs. We use the same finetune sweep for all the pre-trained models in this paper.

Table 4: Finetune details for the pre-trained models.

| Dataset | Learning rate | Total, warmup steps |
|---|---|---|
| ImageNet-1k | $\{0.01, 0.03\}$ | $\{(20\text{k}, 500)\}$ |
| Pets37 | $\{1e\text{-}3, 3e\text{-}3, 0.01\}$ | $\{(500, 100), (2.5\text{k}, 200)\}$ |
| Kitti-distance | $\{1e\text{-}3, 3e\text{-}3, 0.01\}$ | $\{(500, 100), (2.5\text{k}, 200)\}$ |
| CIFAR-100 | $\{1e\text{-}3, 3e\text{-}3, 0.01\}$ | $\{(2.5\text{k}, 200), (10\text{k}, 500)\}$ |
| Resisc45 | $\{1e\text{-}3, 3e\text{-}3, 0.01\}$ | $\{(2.5\text{k}, 200), (10\text{k}, 500)\}$ |

---

[4] As opposed to 3.3 where we specify weight decay values as typically defined in common frameworks, here the values are "decoupled" following (27) that is better suited for sweeps; multiplying weight decay by the base learning-rate recovers the "coupled" value as used elsewhere.

Table 5: Detailed VTAB results, including the "Mean" accuracy shown in Figure 3. We show datasets under ●NATURAL, ●SPECIALIZED, ●STRUCTURED groups, following (45).

| | | Caltech101 | CIFAR-100 | DTD | Flowers102 | Pets | Sun397 | SVHN | Mean | Camelyon | EuroSAT | Resisc45 | Retinopathy | Mean | Clevr-Count | Clevr-Dist | DMLab | dSpr-Loc | dSpr-Ori | KITTI-Dist | sNORB-Azim | sNORB-Elev | Mean |
|---|---|---|---|---|---|---|---|---|---|---|---|---|---|---|---|---|---|---|---|---|---|---|---|
| | | ● | ● | ● | ● | ● | ● | ● | ● | ● | ● | ● | ● | ● | ● | ● | ● | ● | ● | ● | ● | ● | ● |
| **ImageNet-1k (300ep)** | R+Ti/16 | 91.6 | 81.9 | 68.0 | 94.0 | 91.9 | 70.6 | 95.6 | 84.8 | 85.2 | 98.4 | 94.8 | 80.4 | 89.7 | 96.1 | 89.8 | 67.4 | 99.9 | 86.9 | 81.9 | 25.1 | 46.3 | 74.2 |
| | S/32 | 92.7 | 86.4 | 70.7 | 93.6 | 91.2 | 72.9 | 95.8 | 86.2 | 83.6 | 98.6 | 95.5 | 79.6 | 89.3 | 94.2 | 88.4 | 65.8 | 99.9 | 86.1 | 80.7 | 24.9 | 68.2 | 76.0 |
| | B/32 | 92.6 | 87.6 | 72.7 | 94.4 | 92.2 | 73.8 | 95.8 | 87.0 | 82.7 | 98.6 | 94.9 | 79.8 | 89.0 | 94.0 | 89.6 | 66.1 | 99.8 | 84.7 | 80.3 | 24.7 | 62.4 | 75.2 |
| | Ti/16 | 92.7 | 84.0 | 68.9 | 93.8 | 92.5 | 72.0 | 96.1 | 85.7 | 83.7 | 98.7 | 95.6 | 81.6 | 89.9 | 98.0 | 91.9 | 68.5 | 99.7 | 83.2 | 82.0 | 26.5 | 65.9 | 77.0 |
| | R26+S/32 | 90.2 | 86.2 | 74.0 | 95.5 | 94.3 | 74.5 | 95.6 | 87.2 | 84.5 | 98.6 | 96.0 | 83.4 | 90.6 | 99.7 | 91.6 | 73.3 | 100 | 84.8 | 84.5 | 28.2 | 51.3 | 76.7 |
| | S/16 | 93.1 | 86.9 | 72.8 | 95.7 | 93.8 | 74.3 | 96.2 | 87.5 | 84.1 | 98.7 | 95.9 | 82.7 | 90.3 | 98.7 | 91.5 | 69.8 | 100 | 84.3 | 79.6 | 27.3 | 58.0 | 76.1 |
| | R50+L/32 | 90.7 | 88.1 | 73.7 | 95.4 | 93.5 | 75.6 | 95.9 | 87.6 | 85.8 | 98.4 | 95.4 | 83.1 | 90.7 | 99.8 | 90.4 | 71.1 | 100 | 87.5 | 82.4 | 23.5 | 53.0 | 76.0 |
| | B/16 | 93.0 | 87.8 | 72.4 | 96.0 | 94.5 | 75.3 | 96.1 | 87.9 | 85.1 | 98.9 | 95.7 | 82.5 | 90.5 | 98.1 | 91.8 | 69.5 | 99.9 | 84.5 | 84.0 | 25.9 | 53.9 | 76.0 |
| | L/16 | 91.0 | 86.2 | 69.5 | 91.4 | 93.0 | 75.3 | 94.9 | 85.9 | 81.0 | 98.7 | 93.8 | 81.6 | 88.8 | 94.3 | 88.3 | 63.9 | 98.5 | 85.1 | 81.3 | 25.3 | 51.2 | 73.5 |
| **ImageNet-21k (30ep)** | R+Ti/16 | 92.4 | 82.7 | 69.5 | 98.7 | 88.0 | 72.4 | 95.1 | 85.6 | 83.6 | 98.8 | 94.9 | 80.7 | 89.5 | 95.7 | 90.2 | 66.6 | 99.9 | 87.0 | 80.3 | 24.4 | 47.0 | 73.9 |
| | S/32 | 92.7 | 88.5 | 72.4 | 98.9 | 90.5 | 75.4 | 95.4 | 87.7 | 83.5 | 98.7 | 95.0 | 79.5 | 89.2 | 94.5 | 89.8 | 64.4 | 99.8 | 87.9 | 81.2 | 24.9 | 57.7 | 75.0 |
| | B/32 | 93.6 | 90.5 | 74.5 | 99.1 | 91.9 | 77.8 | 95.7 | 89.0 | 83.5 | 98.8 | 95.1 | 78.8 | 89.1 | 93.6 | 90.1 | 62.9 | 99.8 | 89.0 | 78.3 | 24.1 | 55.9 | 74.2 |
| | Ti/16 | 93.3 | 85.5 | 72.6 | 99.0 | 90.0 | 74.3 | 95.1 | 87.1 | 85.5 | 98.8 | 95.5 | 81.6 | 90.4 | 97.7 | 91.7 | 67.4 | 99.9 | 83.8 | 81.2 | 26.3 | 55.1 | 75.4 |
| | R26+S/32 | 94.7 | 89.9 | 76.5 | 99.5 | 93.0 | 79.1 | 95.9 | 89.8 | 86.3 | 98.6 | 96.1 | 83.1 | 91.0 | 99.7 | 92.0 | 73.4 | 100 | 88.7 | 84.8 | 26.2 | 53.3 | 77.3 |
| | S/16 | 94.3 | 89.4 | 76.2 | 99.3 | 92.3 | 78.1 | 95.7 | 89.3 | 84.5 | 98.8 | 96.3 | 81.7 | 90.3 | 98.4 | 91.5 | 68.3 | 100 | 86.5 | 82.8 | 25.9 | 52.7 | 75.8 |
| | R50+L/32 | 95.4 | 92.0 | 79.1 | 99.6 | 94.3 | 81.7 | 96.0 | 91.1 | 85.9 | 98.7 | 95.9 | 82.9 | 90.9 | 99.9 | 90.9 | 72.9 | 100 | 86.3 | 82.6 | 25.4 | 57.4 | 76.9 |
| | B/16 | 95.1 | 91.6 | 77.9 | **99.6** | 94.2 | 80.9 | 96.3 | 90.8 | 84.8 | 99.0 | 96.1 | 82.4 | 90.6 | 98.9 | 90.9 | 72.1 | 100 | 88.3 | 83.5 | 26.6 | 69.6 | 78.7 |
| | L/16 | 95.7 | 93.4 | 79.5 | 99.6 | 94.6 | 82.3 | 96.7 | 91.7 | **88.4** | 98.9 | 96.5 | 81.8 | 91.4 | 99.3 | 91.8 | 72.1 | 100 | 88.5 | 83.7 | 25.0 | 62.9 | 77.9 |
| **ImageNet-21k (300ep)** | R+Ti/16 | 93.2 | 85.3 | 71.5 | 99.0 | 90.3 | 74.7 | 95.2 | 87.0 | 85.2 | 98.3 | 95.3 | 81.3 | 90.0 | 95.5 | 90.5 | 67.4 | 99.9 | 87.4 | 78.2 | 24.5 | 45.2 | 73.6 |
| | S/32 | 93.2 | 89.7 | 75.3 | 99.2 | 92.0 | 78.1 | 96.1 | 89.1 | 84.0 | 98.5 | 95.4 | 80.6 | 89.6 | 96.9 | 88.7 | 68.1 | 100 | 91.0 | 79.6 | 26.2 | 55.0 | 75.7 |
| | B/32 | 95.2 | 92.3 | 77.2 | 99.5 | 92.8 | 81.2 | 96.6 | 90.7 | 87.0 | 98.8 | 96.0 | 81.3 | 90.8 | 97.7 | 89.8 | 70.5 | 100 | **92.3** | 82.7 | 25.9 | **83.1** | 80.2 |
| | Ti/16 | 93.7 | 87.2 | 73.1 | 99.2 | 91.0 | 77.3 | 95.7 | 88.2 | 86.0 | 98.5 | 95.8 | 81.9 | 90.6 | 98.3 | 89.7 | 70.8 | 100 | 86.0 | 82.6 | 26.8 | 49.9 | 75.5 |
| | R26+S/32 | 94.8 | 90.9 | 78.9 | 99.5 | 94.1 | 81.3 | 96.7 | 90.9 | 87.5 | 98.7 | 96.4 | **84.2** | **91.7** | 99.9 | **92.4** | **77.0** | 100 | 87.1 | 83.4 | **28.6** | 56.0 | 78.1 |
| | S/16 | 95.2 | 90.8 | 77.8 | 99.6 | 93.2 | 80.6 | 96.6 | 90.5 | 86.7 | 98.8 | 96.4 | 82.9 | 91.2 | 99.1 | 89.8 | 73.9 | 100 | 87.6 | 85.1 | 26.8 | 61.1 | 77.9 |
| | R50+L/32 | 95.7 | 93.9 | **81.6** | 99.5 | 94.9 | **83.6** | 97.1 | **92.3** | 85.8 | 98.7 | 96.7 | 84.2 | 91.3 | **100** | 92.0 | 76.8 | **100** | 87.2 | 85.2 | 26.8 | 61.8 | 78.7 |
| | B/16 | **96.0** | 93.2 | 79.1 | 99.6 | 94.7 | 83.0 | 97.0 | 91.8 | 87.4 | 98.7 | **96.8** | 83.5 | 91.6 | 99.7 | 89.0 | 76.0 | 100 | 86.7 | **85.7** | 28.3 | 68.2 | 79.2 |
| | L/16 | 95.5 | **94.1** | 80.3 | 99.6 | **95.0** | 83.4 | **97.4** | 92.2 | 86.4 | **99.0** | 96.6 | 83.3 | 91.3 | 99.8 | 91.7 | 75.6 | 100 | 90.4 | 84.7 | 27.5 | 76.5 | **80.8** |

## C  VTAB results

In Table 5, we show all the results in percentage for all the models on the full VTAB. We report VTAB score only for the best pre-trained models, selected by their upstream validation accuracy ("recommended checkpoints", see Section 4.5). For VTAB tasks, we sweep over 8 hyper parameters, include four learning rates {0.001, 0.003, 0.01, 0.03} and two schedules {500, 2500} steps. The best run was selected on VTAB validation split.

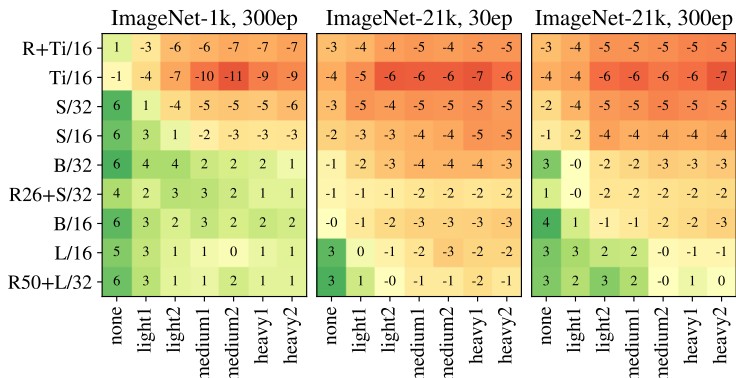

Figure 7: The improvement or deterioration in validation accuracy when using or not using regularization (e.g. dropout and stochastic depth) – positive values when regularization improves accuracy for a given model/augmentation. For absolute values see Figure 4.

## D   The benefit and harm of regularization

In Figure 7, we show the gain (green, positive numbers) or loss (red, negative numbers) in accuracy when adding regularization to the model by means of dropout and stochastic depth. We did verify in earlier experiments that combining both with (peak) drop probability 0.1 is indeed the best setting. What this shows, is that model regularization mainly helps larger models, and only when trained for long. Specifically, for ImageNet-21 pre-training, it hurts all but the largest of models across the board.

## E   Using recommended checkpoints for other computer vision tasks

One limitation of our study is that it focuses mainly on the classification task. However, computer vision is a much broader field, and backbones need to excel at many tasks. While expanding the full study to many more tasks such as detection, segmentation, tracking, and others would be prohibitive, here we take a peek at one further task: multi-modal image-text retrieval.

A detailed analysis of this question is beyond the scope of this study, but we evaluated our *recommended* (see Section 4.5) B/32 checkpoint pre-trained on ImageNet-21k in a contrastive training setup with a locked image tower (46). We initialize the text tower with a BERT-Base (12) checkpoint and train for 20 epochs on CC12M (5). The results in Table 6 indicate that the upstream validation accuracy is a good predictor for zero-shot classification. Moreover, the representations produced by such a model yield similarly better results for image-text retrieval, when compared to models that do not have the ideal amount of AugReg applied. We hope the community will adopt our backbones for other tasks, as already done by (34).

Table 6: Comparing our *recommended* (see Section 4.5) B/32 checkpoint with models that apply too little or too much AugReg. The final validation accuracy from the ImageNet-21k pre-training is the same that is reported in Figure 4. The other columns are ImageNet-1K zero-shot accuracy, and image-text retrieval accuracy on different datasets, after contrastively training as described in (46).

| AugReg | I21k Val | I1k 0shot | Coco I2T | Coco T2I | Flickr I2T | Flickr T2I |
|---|---|---|---|---|---|---|
| none/0.0 | 41.6 | 54.9 | 33.4 | 20.1 | 58.1 | 39.9 |
| heavy2/0.1 | 43.5 | 57.3 | 39.1 | 24.4 | 62.1 | 44.6 |
| Recommended | **47.7** | **60.6** | **41.1** | **25.5** | **65.9** | **46.9** |

