# OpenReview forum: "How to train your ViT? Data, Augmentation, and Regularization in Vision Transformers"
_TMLR — Accepted by TMLR_

### Review · Reviewer_H8zE · 2022-04-20

**Summary Of Contributions:**

This paper presents a comprehensive empirical study on the pre-training and fine-tuning of Vision Transformers (ViT). The aim is to understand how the size of training data, augmentation/regularization techniques, and fine-tuning policies affect the performance of ViT in practical image recognition tasks. The major conclusions include:

(1)	Strong data augmentation and regularization techniques are beneficial, and can alleviate the problem of insufficient pre-training data.

(2)	For many downstream tasks, transferring existing pre-trained models leads to better performance and less computational cost (for training).

(3)	As an efficient model selection policy, one can simply adopt the best pre-trained model on upstream datasets.



**Broader Impact Concerns:**

The authors have summarized the societal impacts and limitations of their work. I do not have some particular concerns on the broader impacts.

**Requested Changes:**

The authors may consider revising this paper according to the listed weaknesses.

**Strengths And Weaknesses:**


Strengths:

(1)	The experiments are pretty comprehensive and well-designed.

(2)	The authors release more than 50 000 ViT models trained under diverse settings on various datasets for further application or research.

Weaknesses:

(1)	The experimental conclusions of this paper seem to be trivial and widely-known, e.g., strong AugRegs helps, large-scale pre-training helps, and fine-tuning is beneficial. Even though in the context of ViT, many recent papers have pointed these out, at least implicitly. Systemically studying these important issues is an acceptable contribution for me. However, I do believe that the authors should comment on this weakness.

(2)	The study is limited to the recognition task and the plain ViT networks. Considering more downstream tasks (e.g., objection detection and semantic segmentation) or the recently proposed pyramid-style ViTs (e.g. swin Transformer and PVT) may make the experiments more strong.

(3)	In Figure 5, if I do not misunderstand, adapting all pre-trained models should always achieve the best performance, since it traverses every pre-trained model. Why there are both positive and negative values in the left Figure?

---

> ### Author Response · Authors · 2022-04-29
> **Reply**
>
> (Please refer to the main "Shared reply for all reviewers" for our reply about extending the study to more ViT variants and tasks.)
>
> We agree that the general theme "AugReg helps" is widely known. We would like to point out though that "strong AugReg" only helps in some cases, and harms in other cases. As we show in Figure 4, the best amount of AugReg depends on model capacity, dataset size, and number of epochs. For example, while strong AugReg increases the validation accuracy from 43 to 51 for the L/16 model when training for 300 epochs on ImageNet-21k, it decreases the validation accuracy from 46 to 40 for the S/16 model on the same dataset, and it would not change the validation accuracy for L/16 when training for only 30 epochs - instead, in that setting applying only "light2" augmentation to L/16 would increase the validation accuracy from 45 to 48. We believe that the value of our study lies on one hand in reporting these differences and reminding the community that the exact effect of AugReg depends on these other parameters, and on the other hand on providing the best vanilla ViT checkpoints pre-trained on ImageNet-21k that can readily be used for a wide range of applications.
>
> As for extending the findings to more tasks and ViT variants, please refer to "Shared reply for all reviewers" above.
>
> Indeed, Figure 5 shows that choosing the best model by validation accuracy does not always select the model with the best test accuracy. For most datasets the effect goes in both directions and is due to some random fluctuation of the last measured score on the validation/test set (that effect is surprisingly large for the Resisc45 dataset). The case of ImageNet-1k is special, because there is a *data leak*: ImageNet-21k data contains ImageNet-1k minival subset. Thus, selecting a finetuned ImageNet-1k model based on minival will favor the models that memorized upstream data well and not necessarily generalize the best. Figure 5 further confirms this, so we use a different set of ImageNet-like images (ImageNet-v2) for model selection.

---

### Review · Reviewer_bnNv · 2022-04-22

**Summary Of Contributions:**

This paper presents evaluations from a hyperparameter sweep on ViT and Resnet+ViT architectures, and evaluates the results to make some conclusions on what matters for downstream performance.

They consider training on ImageNet and ImageNet-21k data, with and without dropout and stochastic depth, with varying amounts of augmentation, and varying amounts of weight decay.

They arrive at a few interesting results. One that jumps out at me is: a model trained with data augmentation performs similar to a model trained without augmentations on 10x data. Another result I like is: there is clearly a huge benefit to transferring models learned on larger datasets (which is maybe obvious), but there is still substantial variance: many transferred models are worse than models trained from scratch. Another interesting result is that regularization hurts the smaller models, and improves the larger ones, whereas augmentation helps them all.


**Broader Impact Concerns:**

All good

**Requested Changes:**

I would like to see a new section, with a title like "Summary of recommendations". Even if these have some caveats, that's OK. I want to know what the authors learned.

**Strengths And Weaknesses:**

The main strength of this work is that it provides a large set of apples-to-apples comparisons, from which people can either take the authors' conclusions or draw their own. The experimental setup appears to be clean and simple, and the results are presented clearly.

A weakness here is that the paper does not use the conclusions to create something entirely new. For example, I might expect that the authors implementing and evaluating all this might walk away with a feeling that they can design a better transformer, a better stem, or a more general or safer strategy for transfer. There are helpful takeaways in the individual sections, but I think I would appreciate something more global. For example, even providing a summary of the optimal choices across the sweep, summarizing the analysis across the many experiments, would be useful.

---

> ### Author Response · Authors · 2022-04-29
> **Reply**
>
> We have added a new section "Summary of recommendations" that highlights the most salient findings in a short itemized list. Please also refer to "Shared reply for all reviewers" above for general remarks and details on other updates of the submission.

---

> > ### Comment · Reviewer_bnNv · 2022-05-03
> > **Better**
> >
> > In terms of readability, I think the third bullet is a little rough ("As for the selection," -- selection of what?). Also the list is inconsistent: the first bullet is a clear recommendation (good), the second is a result (ok), and the third one does not make sense without having read the rest of the paper (bad). I am imagining that a lot of people will want to scroll down to the bottom and read this part alone, and then go back to sections where they want more detail. At least, that's what I sometimes do in a paper like this. So, I recommend putting more effort into clarity here.

---

> > > ### Author Response · Authors · 2022-05-03
> > > **Updated section "Summary of recommendations"**
> > >
> > > Thank you for the quick reply and the suggestions on how to improve the recommendations. We have rewritten the third recommendations to be about which checkpoint to use from our published checkpoints, and we have added the most important references to sections and figures in every recommendation to make it more useful for readers who directly start with the recommendations.

---

### Review · Reviewer_Z7Wx · 2022-04-23

**Summary Of Contributions:**

This paper investigates the training strategies of vision transformers systematically, especially the impact of training data, augmentation, and regularization. There are extensive experiments to get solid findings on vision transformers.

**Requested Changes:**

- The authors conduct most of the experiments on the standard ViT [1]. Could the findings in this paper generalize to other variants of vision transformers?

- Obtaining the conclusions in these papers usually consumes extremely many computing resources, which is usually unaffordable in most of the research institutions. Could toy experiments with fewer training epochs obtain consistent conclusions?

[1] An image is worth 16x16 words: Transformers for image recognition at scale. ICLR 2021.

Overall, I recommend accepting with minor revision.

**Strengths And Weaknesses:**

- This paper presents some meaningful conclusions, which may inspire the community. These findings answer some important questions, whether transfer learning always helps, whether more data enhance the generalization ability, etc. These findings can help researchers/engineers implement a deep learning model to solve practical problems.

- The authors conduct extensive experiments to obtain solid conclusions. For example, the authors investigate 28 configurations of training strategies with different weight decay, data augmentation, and regularization. All these models have been trained adequately with a large lot of computing resources, making the conclusions reliable.

- The experimental results are impressive. With proper training strategy, the model trained on ImageNet-21k can similar performance to those trained on JFT-300M, though there is a large gap between the numbers of the training dataset.

---

> ### Author Response · Authors · 2022-04-29
> **Reply**
>
> (Please refer to the main "Shared reply for all reviewers" for our reply about extending the study to more ViT variants.)
>
> It is an interesting idea to try to find answers to our empirical question about ideal AugReg settings with smaller toy experiments (smaller models, fewer epochs). Unfortunately we believe that this is not feasible, given the insights we have gained from our large sweeps that the optimal strength of AugReg is different in diverse settings. From Figure 4, we see that the smallest models (e.g. Ti/16) have similar patterns when trained for 30 or 300 epochs on ImageNet-21k. But already modestly sized models (in terms of compute, not parameters) like B/32 behave completely differently when trained for 30 epochs (best upstream validation accuracy without AugReg) vs. 300 epochs (worst upstream validation accuracy without AugReg).

---

### Review · Reviewer_Y8hX · 2022-04-24

**Summary Of Contributions:**

The paper conducts extensive experiments, e.g., training 50,000 models under different settings, to investigate the effects of different training recipes and their combinations for training Vision Transformers.  Specific aspects include regularization, augmentation, model size, and training data size. The authors draw a few conclusions from the aspects of (pre-)training and transfer learning in a strictly controlled setting.
Some conclusions are:
1) the gains from increasing dataset size can be alternatively obtained by carefully selecting image regularization and augmentations.
2) transferring a pre-trained ViT is more efficient and effective compared to training from scratch.
3) using more pre-training data leads to better results on unseen downstream tasks.


**Broader Impact Concerns:**

The broader impact concerns are well addressed in Section 6.

**Requested Changes:**

The authors are suggested to add some discussion on other vision tasks or training paradigms as described above. Overall I recommend accepting this paper according to the policy of TMLR, i.e., the claims are convincing and clearly supported by experiments, and the practitioners working on ViTs could be interested in the findings.

**Strengths And Weaknesses:**

Strengths
- The authors provide a consistent setup and 50,000 ViT models with different training recipes, which could be a valuable model set for model analysis.
- The paper is organized and well written. It’s not easy to organize and present so many experiments intuitively. The well-designed figures make it easy to comprehend the corresponding findings.
- The ViTs have become more and more popular. Thus the findings in the paper could be useful to researchers and practitioners working on ViTs, and provide insights into future research.

Weaknesses:
- All the experiments are conducted in supervised image classification. However, the success and popularity of ViTs are beyond it. Do the findings still hold for other vision tasks, e.g., object detection, and other training paradigms, e.g., self-supervised learning.
- I’m not surprised by the findings in this paper. Most of them are natural and straightforward, or commonly held beliefs. For example, either using more data or using appropriate image AugReg can improve accuracy, pre-training is better than training from scratch, and more pre-training data achieves better downstream results.

---

> ### Author Response · Authors · 2022-04-29
> **Reply**
>
> Please refer to the main "Shared reply for all reviewers" for our reply about extending the study to other vision tasks, and about "surprise in the results". Similarly, extending the study to self-supervised learning would have multiplied the resource usage, and added more hyper parameters that would need to be swept for a complete comparison. Instead, in the newly added section in the appendix we adopted a new training/tuning paradigm in our study, i.e. contrastively tune our models for zero-shot transfer, without expanding the scope significantly.

---

### Comment · Action_Editors · 2022-04-25
**Rebuttal**

Dear all:

Thank the authors for submitting the paper and reviewers for providing careful reviews.

According to the comments, this paper is overall good and solid. The main concerns concentrate on: 1. more experiments on other vision tasks expect classification, such as detection and segmentation; 2. evaluations and generalization on VIT-variants (e.g., swin transformer and PVT); 3. more discussions and analysis and on the experiments (e.g., a general strategy for transfer and a comment on utilizing strong aug/regs) .

Can the authors try to provide rebuttals as soon as possible? Thanks.

Regards,
Yunhe

---

> ### Author Response · Authors · 2022-04-29
> **Shared reply for all reviewers**
>
> We thank the reviewers for their comments and suggestions. We're glad to see that reviewers valued the extensive nature of our study and the publication of the checkpoints so the community can build on top of the considerable computational resources that we invested in examining the effects of AugReg on ViT pre-training.
>
> Replies to the main concerns summarized above:
>
> 1. More experiments on other vision tasks: We rely on classification performance as a proxy of the quality of the pre-trained checkpoints. Unfortunately, no clear best setup has emerged yet for object detection/segmentation with vanilla ViTs, and adopting the models with different approaches, i.e. for detection (Chen et al., 2021; Fang et al., 2022; Li et al., 2022), would extend the scope of the study significantly. Instead, we have performed new experiments on one further task: multi-modal image-text retrieval. A small section has been added in the appendix "Using recommended checkpoints for other computer vision tasks", where we contrastively train some of the checkpoints, and report zero-shot classification numbers, as well as image-text retrieval performance.
>
> 2. Evaluation and generalization of ViT variants: As was mentioned by one of the reviewers, the extensive sweep necessary for an apples-to-apples comparison across model sizes and AugReg settings already consumed a great deal of resources. While it would be interesting to extend the comparison to all known variants of vision transformers (e.g., Swin transformer and PVT), this would have multiplied our resource usage, and we believe that the additional insights we would have gained from such a comparison would be smaller and less generally applicable in comparison. It is interesting to note that recent work shows that properly trained vanilla ViT is competitive with more complicated architectures: (Touvron et al., 2022) for classification and  (Chen et al., 2021; Fang et al., 2022; Li et al., 2022) for object detection.
>
> 3. More discussion and analysis: We added a new section "Summary of recommendations" that summarizes the findings from preceding sections for quick reference.
>
> 4. Surprise in the results: we agree that the fact that AuRreg helps to tackle overfitting is known and we do not consider this to be a contribution of our paper. Instead, we carefully quantify the effect of AugReg in a broad and unified study, uncovering a highly delicate and non-trivial landscape, which characterizes the optimal strength of AugReg for diverse settings. We believe this characterization is very valuable for ViT research and was not known before.
>
> References:
> - (Fang et al., 2022) Unleashing Vanilla Vision Transformer with Masked Image Modeling for Object Detection http://arxiv.org/abs/2204.02964
> - (Li et al., 2022) Exploring Plain Vision Transformer Backbones for Object Detection https://arxiv.org/abs/2203.16527
> - (Chen et al., 2021) A Simple Single-Scale Vision Transformer for Object Localization and Instance Segmentation https://arxiv.org/abs/2112.09747
> - (Touvron et al., 2022) DeiT III: Revenge of the ViT http://arxiv.org/abs/2204.07118

---

> > ### Comment · Action_Editors · 2022-04-29
> > **Rebuttal**
> >
> > Thanks the authors for the timely reply.
> >
> > Dear reviewers,
> >
> > Please carefully check the author response to find whether your concerns have been addressed.
> >
> > Regards, Yunhe

---

> > ### Author Response · Authors · 2022-05-03
> > **Updated section "Summary of recommendations"**
> >
> > In response to a comment from reviewer bnNv, we have updated the added section "Summary of recommendations" some more.

---

### Decision · Action_Editors · 2022-05-09

**Recommendation:** Accept as is

**Comment:**

This paper investigates the size of data, augmentation, and regularization techniques in training vision transformers comprehensively. All the reviewers agree that this paper provide useful knowledge and contributions. The main concerns raised by the reviewers is that more experiments and discussions should be provided.

During the rebuttal period, the authors address these concerns by performing new experiments on multi-modal image-text retrieval and adding the "Summary of recommendations" section to summarize the findings across the experiments. All the reviewers give positive recommendations after reading the authors' response.

Therefore, I recommend acceptance for this paper.

---

> ### Author Response · Authors · 2022-05-17
> **Camera ready revision uploaded**
>
> We've uploaded a de-anonymized camera ready revision.
>
> (In this version, we use 05/2020 as publication date, but we can quickly upload a new version with an updated date/year if that's needed.)